

# High resolution data reveal a surge of biomass loss from temperate and Atlantic pine forests, seizing the 2022 fire season distinctiveness in France

Lilian Vallet[1,2], Martin Schwartz[3], Philippe Ciais[3], Dave van Wees[4], Aurelien de Truchis[5], Florent Mouillot[1]

[1]UMR CEFE, IRD, CNRS, Univ. Montpellier, EPHE, 1919 Route de Mende 34293 Montpellier Cedex 5, France (LV : https://orcid.org/0000-0001-6419-1318, FM : https://orcid.org/0000-0002-6548-4830)
[2]French Environment and Energy Management Agency 20, avenue du Grésillé- BP 90406 49004 Angers Cedex 01 France
[3]Laboratoire des Sciences du Climat et de l'Environnement, LSCE/IPSL, CEA-CNRS-UVSQ, Université Paris-Saclay, 91191 Gif-sur-Yvette, France (MS : https://orcid.org/0000-0003-4038-9068, PC : https://orcid.org/0000-0001-8560-4943)
[4]Department of Earth Sciences, Vrije Universiteit, Amsterdam, 1081 HV, the Netherlands (https://orcid.org/0000-0001-5565-7155)
[5]Kayrros SAS, Paris, 75009, France

*Correspondence to*: Lilian VALLET (lilian.vallet@cefe.cnrs.fr, https://orcid.org/0000-0001-6419-1318)

**Abstract.** The frequency and intensity of summer droughts and heat waves in Western Europe have been increasing, raising concerns about the emergence of fire hazard in less fire prone areas. This exposure of old-growth forests hosting unadapted tree species may cause disproportionately large biomass losses compared to those observed in frequently burned Mediterranean ecosystems. Therefore, analyzing fire seasons from the perspective of exposed burned areas alone is insufficient, we must also

consider impacts on biomass loss. In this study, we focus on the exceptional 2022 summer fire season in France and use very high-resolution (10 m) satellite data to calculate the burned area, tree height at the national level, and the subsequent ecological impact based on biomass loss during fires. Our high resolution semi-automated detection estimated 42,520 ha of burned area, compared to the 66,393 ha estimated by the European automated remote sensing detection system (EFFIS), including 48,330 ha actually occurring in forests. We show that Mediterranean forests had a lower biomass loss than in previous years, whereas

there was a drastic increase in burned area and biomass loss over the Atlantic pine forests and temperate forests. High biomass losses in the Atlantic pine forests were driven by the large burned area (28,600 ha in 2022 vs. 494 ha.yr-1 in 2006-2021 period) but mitigated by a low exposed tree biomass mostly located on    intensive management areas. Conversely, biomass loss in temperate forests was abnormally high due to both a 15-fold increase in burned area compared to previous years (3,300 ha in 2022 vs. 216 ha in the 2006-2021 period) and a high tree biomass of the forests which burned. Overall, the biomass loss (i.e.,

wood biomass dry weight) was 0.25 Mt in Mediterranean forests and shrublands, 1.74 Mt in the Atlantic pine forest, and 0.57 Mt in temperate forests, amounting to a total loss of 2.553 Mt, equivalent to a 17% increase of the average natural mortality of all French forests, as reported by the national inventory. A comparison of biomass loss between our estimates and global biomass / burned areas data indicates that higher resolution improves the identification of small fire patches, reduces the





commission errors with a more accurate delineation of the perimeter of each fire, and increases the biomass affected. This
study paves the way for the development of low-latency, high-accuracy assessment of biomass losses and fire patch contours
to deliver a more informative impact-based characterization of each fire year.

## 1 Introduction

Wildfires are a recurrent disturbance across Europe, with an average annual burned area of 475,000 ha since 1980 (European
Commission. Joint Research Center., 2022). This represents on average 0.05% of wildland area each year, with the majority
(95%) occurring in Mediterranean ecosystems which experience drier and warmer summers. In contrast, temperate forests in
Europe are less prone to fires due to their humid and mild climate conditions, and consequently, receiving less attention by the
fire science community (Zin et al., 2022). However, the 2022 fire season in Europe was particularly out-of-the-norm compared
to previous decades, characterized by a severe drought and heatwaves leading to numerous extreme fire events and widespread
burned areas throughout the western part of the continent (Rodrigues et al., 2023). For instance, in Spain, the burned area in
2022 was estimated to be over 300,000 hectares compared to an average of 64,000 ha.yr-1 over the 2008-2021 period according
to the European Forest Fire Information System (EFFIS). The impacts on local populations and firefighting capacities were
significant, drawing the media attention. While the way society perceives these fires and their economic impact on
infrastructures and populations are crucial, it is also necessary to accurately evaluate their immediate ecological impacts to
provide valuable information to societies and stakeholders. This evaluation could reveal unexplored aspects, potentially
challenging the characterization of distinctiveness granted to extreme fire seasons as 2022 solely based on burned areas, which
may overstate or oversimplify wildfire issues "to garner attention in a competitive media ecosystem" (Jones et al., 2022).
In 2022, France has indeed experienced an unusual surge in wildfires during the summer, reaching a burned area of 66,393 ha,
according to automated remote sensing estimates from EFFIS when considering all fire types. This represents a significant
increase from the average of 10,900 ha.yr-1 recorded over the 2008-2021 period by the same source. Notably, the burned areas
shifted to regions outside the traditionally most fire-prone Mediterranean part of the country. The French forests can actually
be divided into three main regions based on their historical fire regimes (Barbero et al., 2019) : (i) Mediterranean forests and
shrublands in the southeast, frequently exposed to fires. (ii) Atlantic maritime pine forests in the southwest, affected by
infrequent but large fires in recent decades, representing an epicenter of large fires in the middle of the 20th Century. (iii) The
rest of the French territory is predominantly agricultural and hosts temperate forests with varying management intensities.
Although this part of France is typically much less affected by fires due to its wetter climate and high landscape fragmentation,
several fire events occurred during the summer of 2022, raising concerns about their environmental consequences.
In recent decades, studies that qualify disturbance factors like fire has increasingly accounted for the ecological impact. Rather
than solely focusing on the extent of the disturbance, a more scientifically-based holistic vulnerability framework, which
combines the concepts of ecological loss and resilience, is employed (Forzieri et al., 2021; Arrogante-Funes et al., 2022). The
vulnerability of an ecosystem, in conjunction with its recovery rate, is highly dependent on the loss experienced by the system





during the disturbance, assessed by its pre-fire state. As forest ecosystems contribute to several ecosystem services (Ninan and Inoue, 2014; Mori et al., 2017), including regulating (carbon sequestration), provisioning (timber and non-timber products), cultural (recreational, aesthetic) and supporting (decomposition, nutrient cycling) purposes, defining a pre-fire state is topic-specific. Tree aboveground biomass is a crucial spatial variable used to evaluate the impact of fires and the resilience of
ecosystem services to fires (Díaz et al., 2018; Martínez-Batlle, 2022; Powell et al., 2014; Sirin et al., 2021; Tyukavina et al., 2022; Volkova and Weston, 2015; Wu et al., 2020), as it serves as a proxy for wood resources and habitat for wildlife and biodiversity (Fusco et al., 2021; Basile et al., 2021; Cazzolla Gatti et al., 2017).

The accurate estimation of the aboveground biomass loss (AGB-L) by fires is challenging due to its spatio-temporal variability, which requires high-resolution data on burned areas and the corresponding spatially varying biomass within each fire patch.
These two pieces of information are yet not available in an operational near real-time impact assessment tool, although recently initiated for Amazonia by Andela et al. (2022). However, they could constitute a keystone knowledge for an accurate comparison of the AGB lost across forest regions and fire events. For instance, a large fire affecting a low biomass plantation may have less impact than a small fire burning an old-growth forest. The challenge is therefore to combine data on the location of burned areas and the impacted forest biomass. Until recently, combining coarse estimates of burned area with the mean
forest biomass of a region was the standard method to assess AGB losses when remote sensing information was not available (Chiriacò et al., 2013; Leenhouts, 1998). Statistical distribution (Prichard et al., 2019) or spatial interpolations (Keith et al., 2014) of biomass using plot data from forest inventories could bring an improved description of the spatial variation of biomass losses, but still lack precise locations to be crossed with actual burned area location. Integrative models of fire emissions combining burned area datasets, land cover, seasonal ecosystem functioning, and a simulation of the biomass carbon pools
affected by fires have been applied globally at 0.5° resolution, in the GFED model (Randerson et al., 2017). A finer resolution of 500 m was recently achieved with this method and constitutes a key information for global studies (van Wees et al., 2022). However, this resolution is still too coarse to capture small fires in Europe. Recent advances in satellite imagery now allow for finer resolution in the burned area detection down to 10 m with Sentinel-2 (Roteta et al., 2019). In parallel, a high-resolution description of forest height and biomass can be obtained from refined land cover and new space-borne Lidar observations of
tree height and canopy structure, as pioneered e.g., in new global maps of forest height at a 30 m resolution (Potapov et al., 2021a). Combining very high resolution and high accuracy maps of burned area and biomass thus opens new perspectives for assessing AGB loss over large areas. High-resolution 20-m burned area detection using the Sentinel-2 MSI sensor already demonstrated an 80% increase in the area burned in Africa compared to the 500-m MODIS sensors, but such data remain to be combined with high-resolution biomass data to assess AGB losses (Ramo et al., 2021).

Hence, we propose in this study to combine a high-resolution exhaustive dataset of fire contours from remote sensing and a new map of tree height at a high resolution (10 m) converted to biomass using local forest inventory plot data to assess the biomass loss in French forests during the fire seasons 2020 to 2022. We discuss the uncertainties of our approach and assess the benefits of high-resolution burned area, and high-resolution biomass maps compared to existing approaches obtained at



coarser resolution. By adopting a multifaceted approach that includes a detailed description of the fire season's distinctiveness,
we will revisit the conclusions drawn from the 2022 fire season in France.

## 2 Materials & Methods

### 2.1 Study Area

In this study, we employed the study of Barbero et al. (2019) and the classification of the French national forest inventory
(NFI) to categorize France into three major regions (Fig. 1):

- The Mediterranean forest and shrubland region (so-called GRECO J and K regions of the NFI (IFN, 2023)) encompasses the
southeast portion of the country and surrounds the Mediterranean basin. This region is composed of low, dense forests that are
dominated by Quercus species (Q. ilex, Q. pubescens, Q. suber) and Pinus halepensis. These species show strategies of
resistance (cork for Q. suber) and tolerance (resprouting for Q. ilex and serotiny for P. halepensis) to cope with the frequent
fire regime that occurs in this region. In addition, sclerophyllous non-forest vegetation, called maquis and guarrigue, is widely

distributed and predominantly affected by fire disturbances (Mouillot et al., 2003).
- The Atlantic Maritime pine forest region (Sylvoecoregion F21 and F22 of the NFI (IFN, 2023)) is almost exclusively
composed of intensively managed forests for timber production. The cultivation of P. pinaster (Maritime pine) has a rotation
time of 20 to 30 years, resulting in a landscape characterized by a mosaic of plots at different growth levels (Petucco and
Andrés-Domenech, 2018; Salas-Gonzalez et al., 2001). This region is less frequently affected by fire than the Mediterranean

region, but fires can spread over large areas and lead to dramatic fire events in the past (Papy, 1950). Despite the serotinous
fire tolerance of maritime pines (i.e., prolonged canopy storage of seeds protected in cones retained on the plant), forest
management practices tend to favor replanting after a patch is affected by fire (Lamont et al., 2020).
- The Temperate forest region corresponds to the rest of the French territory. This zone features a diversity of forest
communities dominated by deciduous and/or coniferous trees and ranging from no to intensive management. Being weakly

affected by fires and due to the absence of a common evolutionary history to this type of disturbance, tree species show a lack
of adapted strategy. Agricultural, pastoral, and other herbaceous vegetation areas comprise a significant portion of this region,
and can be susceptible to prescribed spring fires, such as stubble-burning, particularly in the Pyrenean Mountains area.
Nevertheless, this study will not consider prescribed fires that primarily impact understory or non-forest lands, and that are
largely determined by local decisions rather than climate.

### 125   2.2 Fire polygons - BAMTs method

Due to the lack of reliable and available fire contour dataset over the country, we developed our own fire polygon dataset with
BAMTs (Burned Area Mapping Tools), a semi-automated method of fire contouring at high resolution (Bastarrika et al., 2014;
Roteta et al., 2021). We processed atmospherically corrected and orthorectified images from the L2A product of ESA's
Sentinel-2 mission between 2020 and 2022. The BAMTs method involves the calculation of three spectral indices: Normalized



Differential Vegetation Index (NDVI) (Rouse et al., 1974), Normalized Burn Ratio (NBR) (Key and Benson, 1999), and NBR2
(García and Caselles, 1991). Each fire was first spatially and temporally located using NASA's Fire Information for Resource
Management System (FIRMS) or national registration from the French Base de Données des Incendies de Forêt en France
BDIFF official fire registration (BDIFF, 2023) to target a BAMTs processing zone. Subsequently, we defined the date of
burning to determine a pre- and post-burn period, which enabled us to represent the pre- / post-differences in the 3 indices on

an RGB color scale. From this visual observation, we defined a burned and unburned area, used as a training zone for a random
forest classifier (Belgiu and Drăguț, 2016). This supervised classification was used to detect changes in the NDVI, NBR, and
NBR2 composites. Finally, each produced fire polygon was visually evaluated and manually refined until the desired visual
accuracy was achieved. This process allowed us to remove commission errors (e.g., erroneous fire polygons being detected in
agricultural lands) and reprocess omitted burned areas by enlarging the training zones. A visual inspection follows this semi-

supervised procedure to confirm that the detected perimeters were indeed burned areas. This key step is hardly provided by
automated methods and helps to reach the international standard recommended by the CEOS Working Group on Calibration
and Validation of remote sensing datasets (Franquesa et al., 2020). By focusing on forest fires larger than 30 ha, a total of 113
fire polygons were obtained over the 2020-2022 period of analysis (Fig. 4b shows an example of a BAMTs fire polygon). We
selected only fire polygons that dominantly occurred in forests and shrublands, in turn removing fires occurring on pastures

and grasslands, and thus matching the French forest fire database BDIFF.

### 2.3 Tree height

To obtain canopy height within each fire patch, we followed the methodology described in Schwartz et al. (2022) that yielded
accurate results (MAE = 2.67 m when compared to in-situ forest inventory measurements) over the Les Landes maritime pine
forests in France. The method to map tree height at 10 m resolution in 2020 combines optical (Sentinel-2), SAR (Sentinel-1)

and spaceborne Lidar data (GEDI) with deep learning methods (U-Net model). Sentinel-1 (S1) is a C-band Synthetic Aperture
Radar (SAR) mission launched in 2014 by the European Space Agency (ESA). Here, we used the Ground Range Detected
(GRD) scenes with dual-band cross-polarization (Vertical-Vertical + Vertical-Horizontal bands at 10 m resolution)
preprocessed in Google Earth Engine. We computed a single median composite image of France based on all S1 images from
the leaf-on season (2020-05-01 to 2020-10-01) separated into ascending and descending orbits, thus creating a single composite

image of France with 4 layers: VV_ascending, VH_ascending, VV_descending, and VH_descending at 10 m resolution. The
Sentinel-2 (S2) mission provides multi-spectral images from Earth's surface reflectance with a revisit interval of ~ 5 days.
Here, we used 10 bands of the L2A product (bottom of the atmosphere reflectance) resampled at 10 m when necessary : B2:
Blue, B3: Green, B4: Red, B5-B6-B7: Red edge, B8: Near Infrared (NIR), B8A: "narrow" NIR, B11-B12: Short Wave Infrared
(SWIR). Similarly to S1, we computed the median of all S2 images with less than 50% of clouds for the same time period after

applying a cloud mask.

The Global Ecosystem Dynamics Investigation (GEDI) mission is a spaceborne infrared LiDAR, on the International Space
Station (ISS). It provides energy return waveforms (L1B product) and derived metrics such as canopy relative height (RH)



(L2A product) that describe the vertical forest structure within 25 m diameter circular footprints. In this case, we used RH95 height metrics as reference height in order to train our model. We downloaded all GEDI footprints available for France since the beginning of the GEDI mission and filtered them using the quality flag provided by NASA. U-Net is a fully convolutional network (FCN) widely used in deep learning for image segmentation tasks. Here, we adapted this model for pixel-wise height regression with GEDI data as reference height and S1/S2 images as predictors. With a series of linear operations (convolutions) and non-linear "activation" functions, these types of models can learn multiscale image features, such as image texture, that are then used to carry out height predictions. More specifically, U-Nets (Ronneberger et al., 2015) are divided into a contracting path (left) and an expansive path (right) which gives it its "U" shape and enables the model to extract relevant information at different spatial scales. In total, the network has 18 convolutional layers and ~17 Million trainable weights.

We trained the model following the method described in Schwartz et al. (2022) and obtained a canopy height map for 2020 in France at 10 m resolution (See Fig. 2). Considering that this map has been designed only on the data of the year 2020, we evaluate it to be reliable for the 2020-2022 period only, particularly in the intensively managed Landes region (Petuco and Andres-Domenech 2018), covered by the fast-growing Pinus pinaster species (Serrano-León et al., 2021). Schwartz's innovative high resolution tree height map is the basis for our method of estimating forest aboveground biomass in the remainder of this study. Note that due to GEDI RH95 properties, the heights below ~ 3 m cannot be separated from bare soil or crops.

## 2.4 Aboveground biomass loss (AGB-L)

The workflow in Fig. 3 describes how the Aboveground Biomass losses (AGB-L) were computed for each fire patch, using the 10 m resolution tree height map of Schwartz et al. (2022), the fire polygons obtained with BAMTs, and the French NFI plot data.   AGB-L is defined here as the sum of plant material   released to the atmosphere as gasses and aerosols during combustion, the formation of scorched but surviving trees, of standing and ground-marked dead trees and snags that remain on the ground and may later decompose or be salvaged by forest managers. The calculation of AGB-L at 10 m resolution within each fire patch is performed according to the following successive steps (Fig. 3):

Step 1: Tree height within each fire patch. We cropped the tree height map at 10 m resolution from Schwartz et al. (2022) within the BAMTs fire polygons, keeping only the 10 m pixels with height values higher than 3 m.

Step 2: NFI plots around the fire patch. Data collected since 2005 on more than 100,000 plots were used. The morphological characterization of more than 1.7 million trees allowed the establishment of allometric relationships for each species. For each fire patch, the NFI plots located at a maximum distance of 5 km outside the fire perimeter were selected.

Step 3: DBH ~ height allometry. We established an allometric relationship between the diameter at breast height ($DBH_{NFI}$) and the tree height ($H_{NFI}$) from NFI measurements for each tree of the dominant species in each plot (Eq. (1)). The parameter



p is species-dependent and affects the relationship between height and diameter for the dominant species considered. We computed  p for each fire patch based on all surrounding NFI plots.

$$DBH_{NFI} = \frac{\alpha_p H_{NFI}^2}{\pi} \qquad\qquad 200 \tag{1}$$

Step 4: DBH map. We applied Eq. (2) to the 10 m x 10 m resolution tree height values from Schwartz et al. ($H_x$) to compute the DBH of the highest tree for each 10 m pixel ($DBH_x$).

$$ \tag{2}$$
$$DBH_x = \frac{\alpha_p H_x^2}{\pi}$$

Step 5: Height ~ Biomass allometry. To estimate the above-ground biomass of the highest tree in each pixel ($AGB_x$) from its DBH, we used allometric relationships from the R package allodb (Gonzalez-Akre et al., 2022) for each species. This tool
compares the allometric relations of different studies and builds a new relationship according to the taxonomic and geographical information provided. By considering only the dominant species, we obtained an estimation of the AGB of the highest tree within each 10 m resolution pixel ($AGB_{tx}$) as the result of the allometry function $f_p$ relative to the fire patch, applied to the values of each pixel ($DBH_x$).

$$AGB_{tx} \sim f(DBH_x) \tag{3}$$

Step 6: Tree density estimation. To obtain the aboveground biomass over the whole 10 m x 10 m pixel ($AGB_{fx}$) we need the stand density D which is not provided by remote sensing. We assumed that most of the forests that burned were closed forests,
so that the aboveground biomass of a tree is influenced by the density, according to the "self-thinning" rule commonly used in forestry (Eq 4.) (Yoda, 1963; Puntieri, 1993). Equation (4) relates the forest density at the pixel level ($D_x$) to the biomass of an average tree at the pixel level ($AGB_{tx}$).

$$\log(D_x) = \frac{\log(AGB_{tx}) - k_p}{-\frac{3}{2}} \qquad\qquad 225 \tag{4}$$

In Eq. (4), the parameter $k_p$ has a specific value for each fire patch and was estimated by fitting the equation to all the trees in NFI plots surrounding each fire patch (See appendix N for more details). Finally, we obtained the aboveground biomass of a forest pixel ($AGB_{fx}$) by multiplying the density $D_x$ of the pixel by the biomass of the highest tree of this pixel ($AGB_{tx}$) as
shown in Eq. (5). We assume here that our high resolution 10 m dataset highly limits tree height variability within the pixel,



so that the difference between the average tree height and maximum tree height is below the uncertainty of the maximum tree height (2.75 m) extracted from Schwartz et al. (2022).

$$AGB_{fx} = AGB_{tx} * D_x \qquad\qquad 235 \tag{5}$$

Which led to Eq. (6) expressing the forest biomass of a 10 m x10 m pixel, as a function of the pixel height ($H_x$) and other fire-patch related parameters ($\alpha_p$ and $k_p$) (Chan et al., 2021).

$$AGB_{fx} = f_p\left(\frac{\alpha_p H_x^2}{\pi}\right) * 10^{\frac{2k_p}{3}} \tag{6}$$


Step 7: correction for complex topography. The GEDI-based method tends to overestimate values in areas with complex topography, especially in mountain regions. The slope of each pixel of a fire patch was calculated based on a digital terrain

model (NASA/METI/AIST/Japan Spacesystems And U.S./Japan ASTER Science Team, 2019). Topography was considered complex when the average slope of all the 10 m pixels belonging to the same patch was greater than 3 degrees. In these patches, we used a tree cover mask produced in 2018 (Copernicus Land Monitoring Service, 2023) to remove from the $AGB_{fx}$ estimation all the pixels marked with a tree cover value of zero, where our 10 m AGB values can be considered as unreliable. This correction applies to 12 of the 113 fires considered, mostly located in Corsica and in the Pyrenees.

Step 8: biomass of short sclerophyllous vegetation. The Mediterranean vegetation composed largely of sclerophyllous shrubs (mostly maquis and garrigues) was identified by the CORINE land cover dataset (SDES et al., 2019) as sclerophyllous vegetation class. Within this CORINE class, all burned 10 m pixels with vegetation higher than 3 meters were treated like forests to calculate AGB-L. The pixels with vegetation shorter than 3 meters were considered to be non-forests. In those pixels, our GEDI-based tree height map is not reliable to estimate vegetation height and biomass, because the height of short

vegetation cannot be separated from bare soil. We thus assigned a fixed value of 10t/ha to all those pixels, based on allometric equations of Mediterranean shrublands from De Cáceres et al. (2019).

Step 9: Herbaceous vegetation. Although we focus on forest fires, there can be a fraction of non-woody herbaceous vegetation in each patch, after excluding non-forest sclerophyllous in the previous step. This fraction was estimated to correspond to herbaceous vegetation burning on which was assigned a biomass density of 4tC/ha, based on the national grassland biomass

assessment from Graux et al. (2020).

**2.5 Comparison with other burned area and fire emissions datasets**

The calculation of AGB-L described in Sect. 2.4 was applied on the 113 fire patches from BAMTs. The same calculation has been realized for the whole region outside the burned areas (see Sect. 2.1) to get a rough estimation of living biomass compared to burned one. We also computed AGB-L using two global burned area products and three global height or biomass products



to compare with our high-resolution results (Table 1). The 113 fire polygons from BAMTs were compared, when available, with the national fire polygons estimates from EFFIS (San-Miguel-Ayanz et al., 2012) and MCD64A1 burned area (Key and Benson, 2005; Giglio et al., 2018). Because BAMTs fire dataset focused on forests and shrublands, a large amount of fires and total burned area occurring on pastures and grasslands were removed from the EFFIS and MCD64A1 data. While BAMTs relies exclusively on image analysis from the Sentinel-2 sensors, the EFFIS product relies on both hotspots obtained by the

MODIS sensor and Sentinel-2 images crossover. MCD64A1 (500 m resolution) relies solely on the MODIS sensor.

The AGB-L from each patch from our study was compared with AGB-L calculated from global tree height maps at 30 m resolution delivered globally by Potapov et al. (2021) and processed according to the same method as presented in Sect 2.4 for the Schwartz tree height map. We also used AGB-L directly sampled within each patch from the ESA CCI Biomass dataset at 100 m resolution for the year 2018 (Santoro and Cartus, 2021). It should be noted that this AGB product considers all the

biomass compartments, including foliage and understory vegetation, whereas our AGB product considers only woody tree biomass. Finally, AGB-L obtained in this study was compared to the (burned) wood carbon pool of the global 500-m resolution fire emissions dataset provided by Van Wees et al. (2022), based on the GFED framework and with biomass cycling simulated by the CASA model (van Wees et al., 2022). All these alternative products have a coarser resolution than our burned area and AGB-L maps (Table 1). Overall, we compared our results with an ensemble of three burned areas maps times four AGB maps,

thus 12 different estimates of AGB-L for each of the three forest regions.

## 3 Results

### 3.1 Fire season 2022

During the 2022 fire season, our method estimated a burned area of 42,520 hectares, which is lower than the 48,330 ha estimated by EFFIS for the same selected fires >30ha. Our estimation excluded the spring prescribed fires, leading to a total

of 66,393ha for the whole year as estimated by EFFIS. In addition to the significantly higher burned area, this fire season was characterized by an unusual distribution outside the high fire-prone Mediterranean region. This south-eastern region of France, composed of Mediterranean forest and sclerophyllous short vegetation (shrubland, maquis, and garrigues) has been the most affected by fires each year since 2006. According to BDIFF (2006-2019) and our BAMTS estimates (2020-2022), this region (blue area in Fig. 4.a) has an average annual burned area of 4,812 ha.year-1 from 2006 to 2021 (Standard deviation (SD) =

3855 ha) with high fire years observed in 2009 (8,403 ha), 2016 (9,800 ha) and 2017 (15,660 ha). Over the same period, this fire-prone part of the French territory accounted for 73.5% (SD = 18.4%) of the burned surface. Unlike most western European regions, the southern region of France did not experience an extreme fire season in 2022, with a burned area of 7,971 hectares, moderately above the mean, but representing only 18.7% of the total area burned in France that year. In contrast, the rest of the French territory, separated into two types of forests in Fig. 4.a (temperate and Atlantic maritime pine) experienced a

significantly different burned area. The burned area in the maritime pine forest reached 26,858 ha in 2022 (Fig. 4.c) compared to an average of 494 ha.year-1 (SD = 379 ha) over the 2006-2021 period. In 2022, the total number of fires larger than 30 ha



detected was 15, compared to an average of 1 to 5 fire events per year observed during the previous years. In this region, the average burned surface per fire event reached 3,357 hectares in 2022, compared to 216 (SD=229) over the 2006-2021 period. Another keystone result was the abnormally large area of temperate forests that burned in 2022, reaching a total of 7,813 ha,

which is 2.6 times larger than the maximum of all the previous years (2019, burned area = 3,035 ha). The most important temperate forest fire events occurred in Brittany, the Loire valley, and the Jura regions (Fig. 4.a). Although the burned area in this region was significantly higher than the previous years, it was composed of only a few small fires, with an average fire size of 174 ha and a maximum fire size of 1730ha. We conclude here that the 2022 fire season differs significantly from previous years' fire season since 2006 in many aspects, including the total burned area, the fire sizes and the region affected.

Consequently, the impact on biomass loss over the territory cannot be derived from simple correspondence rules based on previous years observations. Therefore, further investigation was conducted on the biomass affected.

### 3.2 Tree height and biomass assessment on fire patches

The accuracy of AGB-L assessment mostly depends on the accuracy of the 10 m tree height map sampled within fire patches. This tree-height product was previously validated in Schwartz et al. (2022) against NFI plot data over the maritime Pine forest

of the Landes, showing a MAE of 2.67 m. Here, we extended this validation effort to other forest regions using a larger set of NFI plot measurements collected during 2019-2021. Only the plot locations within a 5-km area around each fire patch were considered (n = 451). The validation results presented in Fig. 5 show a good agreement between our 10 m resolution canopy height map and the NFI height values, with mean absolute errors of 2.38 m, 2.85 m, and 2.59 m for the Mediterranean (n = 118 plots), Maritime pine (n = 86 plots), and the temperate forests respectively (n = 247 plots). Outliers in Fig. 5 where the

predicted height is close to zero, are likely to be forests that were measured by a former NFI census and were then clear-cut during the observation period for which our height map is established, especially in intensively managed forests such as les Landes (Maritime Pine Forest). Removing these outliers (when the predicted height < 4 m) results in a MAE of 2.08 m.

As a second step, we examined the relationship between tree height in 10 m burned pixels and biomass calculated as described in 3.4 (Fig. 6). We observe that, depending on the region considered, and even within one region, tree height and biomass are

not correlated in the same way. In Mediterranean forests, trees are relatively smaller but biomass increases faster with height due to a high tree density and hardwood species. Conversely, in the Atlantic Maritime Pine forest, trees can reach upper canopy heights up to 25 m but with lower biomass due to high forest management limiting tree density and lower wood density for this species. Interestingly, for the old unmanaged forest of La Teste de Buch (tree height ~25 m), we observe a different tree height/biomass correlation, with a higher biomass of highest trees compared to the neighboring managed forest. Temperate

forests show an intermediate tree height/biomass relationship with more biomass at a given tree height than in the Maritime Pine forest but less than the Mediterranean forests, with a bimodal relationship between needle leaf low wood density and hardwood forest with high wood density co-occurring in the region (Fig. 6).



### 3.3 Aboveground biomass loss by fires

When crossing our fire patches with our biomass map, our results reveal a marked increase in the amount of aboveground
biomass loss (AGB-L) between the 2022 fire season (2.553 Mt) and the two preceding years (2020: 0.361 Mt, 2021: 0.805
Mt). Although the burned area in 2022 was significantly larger than the two previous years, being 10.3 and 4.5 times larger
than in 2020 and 2021, the increase in biomass loss was proportionally lower, being only 7.1 and 3.2 times larger, respectively.
Between 2020 and 2021, the increase in biomass loss (times 2.21) was almost proportional to the burned area increase (times
2.38).

The 2022 fire season showed a major contribution of biomass loss from the maritime pine forests in Les Landes (68.2%; 1.74
Mt), and by temperate forests (22.3%; 0.57 Mt), associated to a lower relative contribution of 63.0% and 18.3% in burned area.
The lower impact of fires on biomass loss in 2022 compared to the burned area affected is mainly due to the low contribution
of biomass loss observed over Mediterranean forests, which represents only 9.8% (0.25 Mt) of the total loss during the year
2022. In this latter region, the AGB-L was surprisingly 74% lower than in 2021 and 42% lower than in 2020. This decrease in
ABG-L cannot be attributed to a reduction in burned area alone, because in 2022 (7971 ha) about as much area was burned as
in 2021 (7275 ha) and more than in 2020 (3141 ha). This difference in the Mediterranean biomass / burned area balance
requires a more comprehensive analysis of biomass distribution within regions.

### 3.4 Biomass loss distribution in burned areas

To investigate how the spatial heterogeneity of biomass within and across fire patches affects AGB-L, we conducted a study
on the biomass distribution within burned areas. The distribution of the AGB-L density (t/ha) over burned areas during the
period 2020-2022 is shown in Fig. 8. In 2022, the distribution of AGB-L in Mediterranean forests has a left-hand skewed
distribution, with a mode at 10 t/ha. This value corresponds to the sclerophyllous biomass density that we fixed and could not
be derived from tree height data. This burned biomass distribution is actually different from the available biomass distribution
(red in Fig. 8) over the whole Mediterranean region. Our findings indicate that the 2022 fires in the Mediterranean region
primarily impacted low biomass vegetation, with 88.6% of the burned area affecting biomass values under 100t/ha. Notably,
sclerophyllous short vegetation, which represents 26% of the woody land cover, was found to be proportionally much more
affected by fires (fig 8.a for years 2020 and 2021). This vegetation type has a biomass of only 14% of the median biomass of
the entire region. Consequently, the higher propensity of shrublands to fire reduces the impact on biomass loss. This skewed
distribution toward low biomass vegetation was even more prevalent in 2022 than during previous years.
Our study reveals that the distribution of AGB-L in Maritime pine forests exposed to fire in 2022 shows a binomial distribution.
The first and main peak from 20 to 80t/ha (35.5% of the distribution) in Fig. 8.b corresponds to the intensively managed mono-
specific stands with heterogeneous ages. The area occupied by young stands with a low biomass is larger than in other regions
due to shorter rotations and a larger harvested fraction. Notably, the main part of the burned distribution is actually close to
the biomass distribution across the entire region. The second peak occurs over a very high biomass density (around 270 t/ha,



10.2% of distribution) representing the old-growth, unmanaged forest that burned during the extreme fire of La Teste on the coast. The La Teste fire AGB-L outlier illustrates one of the distinctive facets of the 2022 fire season.

Furthermore, the distribution of AGB-L in burned temperate forests in 2022 shows a rather homogeneous distribution between 20t/ha and 250t/ha. The high biomass AGB-L range of 150-250t/ha corresponds to 14.3% of the distribution. This distribution of AGB-L contrasts with the distribution over the entire region, indicating that the 2022 fires affected a relatively higher

proportion of old-growth forests than were available to burn. This impact on carbon loss was greater than expected and corresponds to another distinctive facet of the 2022 fire season.

**3.5 Comparison of burned area and biomass loss datasets**

The refined BAMTs method (see Sect. 2.2) show differences in the detection rate and total area of fire polygons compared to EFFIS and MCD64A1 products (Table 2). In contrast to BAMTs, EFFIS detected only 89% (101 fires) of the 113 fires due to

a lower detection capacity of small fires (<100ha), as only 51 of 63 fires were recorded (81%). However, all fires >100ha were detected. Despite its limited capability to detect small fires, EFFIS reported a much higher total burned area of 62,002 ha over the period 2020-2022 compared to 56,053 ha in the BAMTs method. This 10% overestimation by EFFIS was primarily due to the overestimation of large fires (>500ha) with a total burned area of the 16 largest fires of 2020-2022 being 4,649 ha (+10%) higher. On the other hand, MCD64A1 product presents a higher burned area in the same sample of 113 fires detected by

BAMTs, despite its lower detection rate of only 39% (44 fires), which is much lower than EFFIS. Although MCD64A1 was capable of detecting almost all large fires (>500 ha), it was unable to achieve the same accuracy on intermediate fires (100-500ha) with a detection rate of 68% (23 fires) and even less on small fires (<100ha) with a detection capability of only 10% (6 fires). Over the matching fire events, we observe that MCD64A1 tends to overestimate the burned area for fires larger than 500 ha by 16% (Fig. 9).

Regarding our estimate of AGB-L (Sect. 2.4), we found higher values than the one obtained from global products (Fig. 10). When our biomass estimation method is applied to the tree height data from Potapov et al. (2021), we estimate that the AGB-L is 8% lower. This difference reflects the propensity of our 10 m resolution tree height map to capture higher trees and their spatial variability within forest patches, including forest edges. Secondly, although the ESA CCI Biomass data takes into account the fine elements of the trees (branches, leaves), it provides a much lower estimate of AGB-L (-32%). This is

particularly related to the underestimation of the biomass of Mediterranean ecosystems. Finally, the Van Wees et al. (2022) AGB-L seems relatively close to our estimates (-9%) despite a much lower resolution. The major difference also appears in the Mediterranean region, for similar reasons as for ESA biomass CCI. In conclusion, the lower resolution products tend to estimate a higher burned area, but a lower biomass being affected.



## 4 Discussion

### 4.1 Uncertainties in burned area and biomass estimates


Our study represents a significant step towards the development of a national monitoring system for burned area polygons at high resolution for fires larger than 30ha. This type of dataset is not yet available neither for burned areas estimations delivered by forest services and previously used in scientific studies (Ruffault and Mouillot, 2017), nor regional/landscape applications relying on coarse resolution global datasets (Barbero et al., 2019) or hand-drawn fire contours

(Mouillot et al., 2003; Ganteaume and Barbero, 2019). To ensure the reliability of our dataset, we visually checked and adjusted each fire contour following international standards (Franquesa et al., 2020). We also set the seed of observation from registered fire events by french forest services (BDIFF), potentially non-exhaustive, completed by thermal anomalies (this latter source being the only source for 2022, before fire events are published by official statistics) following Majdalani et al. (2022). Our estimate of a forest burned area of 42,520 ha was lower than the official 66,363 ha provided by EFFIS for 2022, as we focused

our study on forest and shrubland fires. This is a major limitation when comparing registered national burned area's statistics with remote sensing data not filtered by land cover, as pointed out by (Turco et al., 2019). EFFIS provides a total burned area assembling all sources of fires without filtering out fire types, an information largely directly used by the media. To ensure consistency with BDIFF, the French fire observation system, we focused on forest or shrubland fires, voluntarily omitting pasture and agricultural fires where uncertainty is still high (Hall et al., 2021). We warn here the use of raw data provided by

remote sensing services without understanding vegetation types affected. When filtering out pastoral fires, covering roughly 15,000ha and occurring between February and April when prescribed burning is allowed in the Pyrenees and Central France, EFFIS detected 48,328 ha, much closer to our estimates, with a 10.6% overestimation, a result actually consistent with Llorens et al. (2021). Mostly, EFFIS tends to produce a smooth external envelope of the burned polygons and ignores unburned patches within the fire patch. Still, EFFIS remains a reliable dataset when this limitation is known. We observed a much greater

uncertainty on burned areas with global remote sensing datasets at coarse spatial resolution (500 m). Mostly, small fires <100ha were not observed by MCD64A1, fires between 100ha and 500ha were underestimated and fires larger than 500ha were overestimated by 10% with the same reason as for EFFIS as they provide an external envelope omitting unburned internal patches. This caveat and the fire size threshold for reliability are consistent with (Nogueira et al., 2016), or (Katagis and Gitas, 2022) and (Galizia et al., 2021) in the Mediterranean area. To prevent misinformation to stakeholders in the future, we suggest

here to increase efforts in developing a long-term nationwide observation system quickly collected and analyzed during emergency events as the 2022 fire season.

Regarding biomass, we estimated higher values than other automated global biomass information by 8% to 32%, mostly as a consequence of varying spatial resolution among datasets, where coarse resolution tends to underestimate higher biomass values as previously reported (Yu et al., 2022). Our AGB-L estimates rely on assumptions on tree allometry and tree density.

We derived AGB only from tree height provided by remote sensing data, while standard allometry equations rely on both three height and diameter at breast height (DBH). Field-based generic allometric equations are acknowledged to be locally variable



(Henry et al., 2015) with some significant, potentially high (up to 97%) uncertainty and impacts on C stock estimates (Vorster et al., 2020). Lidar-based biomass estimates, capturing the whole tree architecture, highly recently improved the quantification of carbon stocks in standing trees (Xu et al., 2021), fuel load for fire hazard (Fares et al., 2017), and $CO_2$ emissions (Domingo

et al., 2017), when compared to multispectral images. This latter method has been mostly based on foliage information as a proxy to tree structure, thus only partially informative in logged areas where crown development saturates while tree height continues to increase (Jubanski et al., 2013). Our method, using tree height of the tallest tree at high resolution, then provides a step forward for a spatially continuous estimation of forest biomass in the highly managed French forests beside the plot-based forest inventory. The uncertainty reached in our method includes the tree height uncertainty from Lidar (Schwartz et al.

2022) with an MAE varying between 2.38 m and 2.85 m for trees height varying between 3 and 30 m, leading to a 10% error. The additional error associated with the conversion of tree height to biomass is a major bottleneck in the current biomass density maps at the European or global level. Saito et al. (2022), for example, pointed out a 35% difference between two reference global biomass datasets (Globbiomass and GEOCARBON) and Avitabile and Camia (2018) report an error reaching up to 58% to 67% in Europe for biomass maps elaborated from multispectral remote sensing between 2000 and 2010, so that

harmonized global ABG map still report a 10 to 26% variation over Europe (Spawn et al., 2020). Using the recent methods of Lidar and GEDI data, Duncanson et al. (2022) report a 40% to 50% RMSE from a generic method of European biomass estimates. They however also report local studies with lower errors, around 13 to 25%. Indeed, for example, Fassnacht et al. (2021) actually reach a 9.1 to 15.7% RMSE for biomass estimates from Lidar in Mediterranean Chile, and Simonson et al. (2016) obtained a 10% error in estimating Mediterranean woodland carbon stock increment from Lidar compared to forest

inventories. Our method then uses the benefits of the fine resolution GEDI data with locally-fitted allometric conversion rules to minimize errors in biomass estimates.

Our method also used the tallest tree height detected from GEDI data as a proxy for the forest structure at the 10 m pixel level, as suggested by (Meyer et al., 2018). This constitutes a potential source of uncertainty in our study, as we had to assume a tree density function decreasing with tree height according to the generic 'self-thinning' rule used in forestry in the absence of any

tree density information. Tree crown delimitation from fine resolution Lidar should deliver new information to capture tree density and then reduce biomass estimate uncertainties in the future (Weinstein et al., 2021). With an average tree crown of 3 m to 5 m diameter, we expect 4 to 9 trees over the 10x10=100m²2 pixels, close to the average 500 trees per hectare observed in France. We expect the high resolution used in our study to reduce the error associated with the tree density function on such a low number of trees. Ultimately, it should be noted that we chose to ignore our biomass estimates for tree heights lower than

3 m, and replaced it by an average shrub and grass biomass from the literature, as Lidar data remain highly uncertain (>100% error) for this forest class of tree height (Urbazaev et al., 2018; Durante et al., 2019). Shrub (Li et al., 2022) and grassland (Graux et al., 2020) biomass, can be more related to foliage proxy detected by Lidar or multispectral images and should be further explored for fire impact assessments. Biomass in the understory vegetation, yet hardly provided by remote sensing (Ferrara et al., 2023), would refine carbon stocks for fire impact or fire hazard assessments from this highly flammable and

fire conductive vegetation layer. We also acknowledge that our study only covers a short period from 2020 to 2022, as we





chose to analyze the fire impact close to the GEDI image acquisition. Tree height information is actually highly dynamic as a result of tree cover change due to disturbances or management plans (Hansen et al., 2013; Senf and Seidl, 2020), particularly in the highly managed Landes region, preventing the use of our database backward in time without increasing bias. Further routinely updated biomass estimates would pave the way for recurrent and accurate impact assessment as we could show the

benefits of quantifying biomass loss at fine resolution for this 2022 fire season.

**4.2 Biomass-based vulnerability assessment redefines the 2022 distinctiveness**

From our refined burned area and AGB-L estimates, we conclude on a total biomass affected by fire of 2.553 Mt in 2022. This

loss of biomass corresponds to 8.94% of the biomass harvested each year (28.5 Mt) and further used for energy (13.5 Mt), construction (9.5 Mt) and industrial (5.5 Mt) sectors (EFESE, 2023). More generally, fires impacted an amount of biomass corresponding to 3.86% of the annual gross forest production (i.e., forest wood growth) which is 66 Mt.y-1. This fire affected biomass then contributed to a 17% increase of the average natural mortality of all French forests, as reported by the national inventory.

We'll note that most of the anomaly in 2022 comes from rarely-observed fires in the Atlantic maritime pine and in temperate forests. The high biomass lost in Les Landes is explained more by the large abnormal burned area. This large burned area had a lower than expected impact on AGB-L in the maritime pine forests because fires affected mostly managed forests with a 20/30 years exploitation return interval. The media actually thoroughly reported images of burnings in the old-growth forest of Teste de Buch with 5 times more biomass, and located behind the emblematic and touristic Pyla sand dune. This old-growth

forest represented however only 10% of the burned area in Les Landes and 30% of the biomass loss.

A yet poorly reported information is the low-burned area in the Mediterranean region in 2022 (Rodrigues et al. 2022). In addition, we could show in our results the low biomass affected (mostly shrublands) compared to all available burnable biomass in the region. Fire return intervals in Mediterranean ecosystems vary between 15-25 years for shrublands and 70-120 years for forests (Mouillot et al., 2002) so that 4 to 5 times more burned area in shrublands is expected, while we experience 6 times

more burned area in shrublands than forests in 2022.

The most concerning impact is then the high biomass lost in temperate forests due to high biomass densities. Unmanaged and low fire-prone temperate forests have been accumulating large amounts of biomass across the last decades and thus were highly impacted when the fire occurred. The burned area remains low and represents only 0.056% of the temperate forest surface. Accounting for biomass loss appears as a substantial informative variable to characterize a fire season and its

ecological impacts, rather than burned area alone, potentially misleading when affecting contrasted ecosystems, sylvoregions, and forest structures.

As a first attempt, we focused on estimating AGB affected by fires without separating surviving trees, dead trees and gaseous / aerosols emissions into the atmosphere. Future development of fire impact assessment should dig further into total biomass,



including foliage and fine fuel such as understory and litter (Li et al., 2022). This biomass fraction is economically less
valuable, but highly combustible and contributes to carbon and pollutants emissions to the atmosphere. Mortality in Quercus
species is actually low, and previous research could show the high mortality rate in pine forests (Garcia-Gonzalo et al., 2011).
The high crown architecture of Pinus pinaster could induce a low combustion rate and lower carbon emissions compared to
other species. Combustion efficiency, defined as the fraction of biomass affected emitted to the atmosphere as $CO_2$, is usually
assumed not to exceed 20% for the woody compartment of forests (Mouillot et al., 2006), with even lower values of 0.5% to
3.5% in Mediterranean forests (Harmon et al., 2022), leaving a large amount of standing biomass for decomposition (Campbell
et al., 2016) or harvesting. Post-disturbance harvesting is actually a dominant strategy in les Landes to collect timber and
biomass for low-value wood products or the paper industry. For 2022, the carbon impact may be minor, while the landscape
value of the touristic Landes forest might have suffered more regarding its attractiveness and emotional aspects to society
(Tribot et al., 2018). One final major, yet hardly, assessed impact might be soil carbon combustion. This neglected aspect in
most ecosystems in Mediterranean Europe due to low carbon stock is a major concern in boreal forests and peatlands (Wiggins
et al., 2021). The soil in les Landes pine forest is actually slowly decomposed due to the chemical properties of the needles
growing on a well drained sandy soil, thus accumulating more carbon in the soil than in Mediterranean forests as suggested by
the upper soil carbon stocks inventory of France (Martin et al., 2021). Rough estimates of forest soil carbon deeper than the
upper 20 cm do not exist at the national or European level, and smoldering combustion is yet hardly detected by remote sensing.
We might suspect high smoldering combustion after the 2022 fire season in les Landes as identified by local firefighters when
the Landiras fire started again in a second large fire from soil combustion lasting more than 2 weeks, a phenomenon previously
observed in Mediterranean pine forests (Xifré-Salvadó et al., 2020). This should be further investigated in terms of soil carbon
loss and combustion impact on atmospheric emissions.

An additional step to further account in the holistic evaluation of fire ecological impacts for the year 2022, would be the
resilience capacities of the exposed species, as Qin et al. (2022) showed how the high fire season 2019-2020 in Australia
rapidly recovered from efficient regenerating strategies of the affected vegetation. Mediterranean shrublands, and Quercus
species are efficient resprouters with high resistance (low combustion or mortality) and rapid recovery rates due to basal or
tree buds regeneration, reducing the ecological impact of the 7,971 ha affecting this region. Pinus pinaster regenerates from
seeds and should recover from this disturbance as the previous large fire in the region occurred more than 15 years before, thus
allowing tree maturity and seed production. In addition, this highly managed forest was rapidly harvested after fire to collect
partially burned stems, and will be rapidly planted for a fast and efficient recovery with a growth rate of 0.2-0.7 m.year-1
(Lemoine, 1991). We might be more concerned about the temperate forests affected in 2022 with poor resistance strategies
and low growth rates. This should be included in a fully holistic vulnerability framework to better drive communication to
societies and political decisions (Forzieri et al., 2021). Post-fire recovery rates in each region could be evaluated from historical
fire polygons and the current biomass map produced in this study for 2020 as performed by Berner et al. (2012) in the Euro-
siberian Boreal forest.



**4.3 The 2022 fire season in regard to historical and future fire regimes**

We focused our study on the 2022 fire season, referenced as extreme for France and the overall European context, and compared this year to the two previous years. An extended comparison to the previous 6 years shows that 2022 was an exceptional fire year compared to preceding years, particularly outside the Mediterranean region. The extreme hot and long drought of 2022 can thus be viewed as a 'natural experiment' to assess the biomass loss by new fire regimes that never or merely happened in the recent past. This conclusion that the fire season of 2022 never happened before was highly reported in the media. However, this conclusion was based on the raw EFFIS dataset, covering only the last 12 years, and including pastoral fires. In the last decades, the Mediterranean region experienced way more total burned area and larger fire sizes, such as in the year 2003 when a heatwave covered most of Western Europe (Ciais et al., 2005), and even earlier in 1989, when fire prevention was less effective than nowadays (Ruffault and Mouillot, 2015). Regarding the Atlantic maritime pine forests, this region is historically less affected than the Mediterranean, but rare large fires actually occurred in the late 1940s, e.g., in 1949 when 130,000 ha were burned (Papy, 1950). We could argue that fire fighting services back in the 1940s were less efficient, but the burned area covered 5 times the burned area of 2022 with climatic conditions close to or less extreme than in 2022. For the temperate forest fires, little information has been registered for France, but the year 1976 was marked by a severe heatwave and drought and experienced large fires in northwestern France (Boulbin, 1978; Jean and Larue, 1999). If 2022 is part of the most extreme years over the recent decades, it cannot be claimed as 'never observed before'. In the absence of any reference historical database for France and other European countries covering a longer period than the remote sensing era (a few decades) misleading short-term views may continue to spread out. We encourage historical data gathering, although incomplete (Pausas, 2004; Koutsias et al., 2013; Mouillot and Field, 2005), to better characterize high fire years as exceptional or unique or novel. Still, the year 2022, with all the information available on biomass affected and precise fire contours (yet hardly available before) could be used as a reference fire year potentially more regularly happening in the future.

Actually, future climate projections show how warmer temperatures combined with more frequent and more extreme drought in Europe could potentially increase the risk of fires in the future (Mouillot et al., 2002; Ruffault et al., 2020; Fargeon et al., 2020). As a result, fire indicators such as the Fire Weather Index (FWI) are expected to increase strongly in the coming decades when considering scenarios where carbon emissions are constantly increasing (RCP 8.5) (Fig. 11). The high average seasonal FWI values currently observed in the Mediterranean region of France (26.75 in 1970-2005) are expected to continue to increase until reaching an extreme danger level during the whole summer period (38.7 in 2080-2097). Despite a lower-than-usual mean FWI level in this region in 2022 compared to the previous decade, the burned area was higher than normal (7,971ha compared to 4,812 ha.year-1 in 2006-2021) and could tend to increase with a prolonged fire season and repeated heat waves. In the maritime pine forest, the average seasonal FWI is expected to almost double, from an average of 11.23 over the period 1970-2005 to 20.24 over the period 2070-2097. The year 2022 was marked by an average FWI value in the normal range (12.69) of the previous decade but reaching very high local extremes (95th percentile = 18.75). This high margin of the distribution of FWI, closely linked to the extremely large fires this year, is projected to become the norm in the years 2050-2080 and could



even become the low margin in the period 2080-2097. The temperate forest experienced abnormal FWI in 2022. Projections estimate that the mean seasonal FWI should more than double between the historical period (8.59 in 1970-2005) and the end of the century (18.01 in 2080-2097). 2022, with an average seasonal FWI value of 13.46, already gives us a glimpse of what may be the norm in the years 2050-2070. But the fire risk is expected to be even higher at the end of the century, exceeding the values obtained for this year, which was marked by many large fires in the North of the country. Impacts on fire-induced

CO2 emissions might in addition shift toward more emitting fire events (Carnicer et al., 2022).

**Authors contribution**

LV and MS performed data curation and analysis and wrote the manuscript. FM and PC wrote the manuscript and supervised the project. AT and DVW revised the manuscript. DVW provided GFED-framework data.

**Acknowledgements**

This work was supported by the French Environment and Energy Management Agency (ADEME) and the FirEUrisk H2020 project. FirEUrisk project has been granted funding from the European Union's Horizon 2020 research and innovation programme under the Grant Agreement no. 101003890. This work was also supported by the Climate Change Initiative (CCI) Fire_cci Project (contract 4000126706/19/I-NB).

Fine resolution long-term burned area observation from remote sensing over France is supported by OSU OREME. Fire data

and biomass estimation will be available upon request on https://oreme.org/observation/foret/incendies/.

We are also grateful to Cedric Vega (IGN) who made possible the validation of our height map with the French NFI plot data.

**Data and code availability**

Fire polygons and their AGB-L assessments is available upon request to the corresponding author and through the OSU ORME website (https://oreme.org/observation/foret/incendies/)

Fire inventory data can be downloaded from The French National Inventory platform (https://inventaire-forestier.ign.fr/dataifn/)

Other Fire polygon data can be downloaded from the Copernicus EFFIS website (https://effis.jrc.ec.europa.eu/applications/data-and-services) and ffrom NASA MCD64A1 website (https://lpdaac.usgs.gov/products/mcd64a1v006/)

Other biomass data can be accessed through GLAD website for Potapov tree height (https://glad.umd.edu/dataset/gedi), ESA CCI Biomass webpage (https://climate.esa.int/en/projects/biomass/) and supplementary materials of Van wees et al.(2022)



**Competing interests**

The authors declare that they have no conflict of interest.

**Appendix**

**1 Density estimation of a NFI plot**

In French NFI plot data, trees are categorized into three wood diameter classes : small wood (s) between 7.5 cm and 27.5 cm excluded, medium wood (m) between 27.5 cm and 47.5 cm excluded and large wood (l) greater than 47.5 cm. The enumeration (n) of each tree is carried out according to three circumscribed circles as follows: (1) in radius of 6m, all the small (ns1), medium (nm1) and large (nl1) woods, (2) in radius of 9 m only the medium (nm2) and large (nl2) woods and (3) in radius of

15 m, only the large woods (nl3). Thus small woods are sampled on an area of As=6²113.10 m², medium woods on an area of Am=9²254.47 m² and large woods on an area of Al=15²706.86 m². To calculate the stand density of the plot (D, in nb/m²) we summed of the density of each diameter class as follows :

D=ns1As+nm1+nm2Am+nl1+nl2+nl3Al

We then used this density D, along with the AGB of each tree calculated with the allodb package in order to fit the 3/2 self-

thinning rule and find the coefficient k. described in 3.3.3.

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



**Figures**

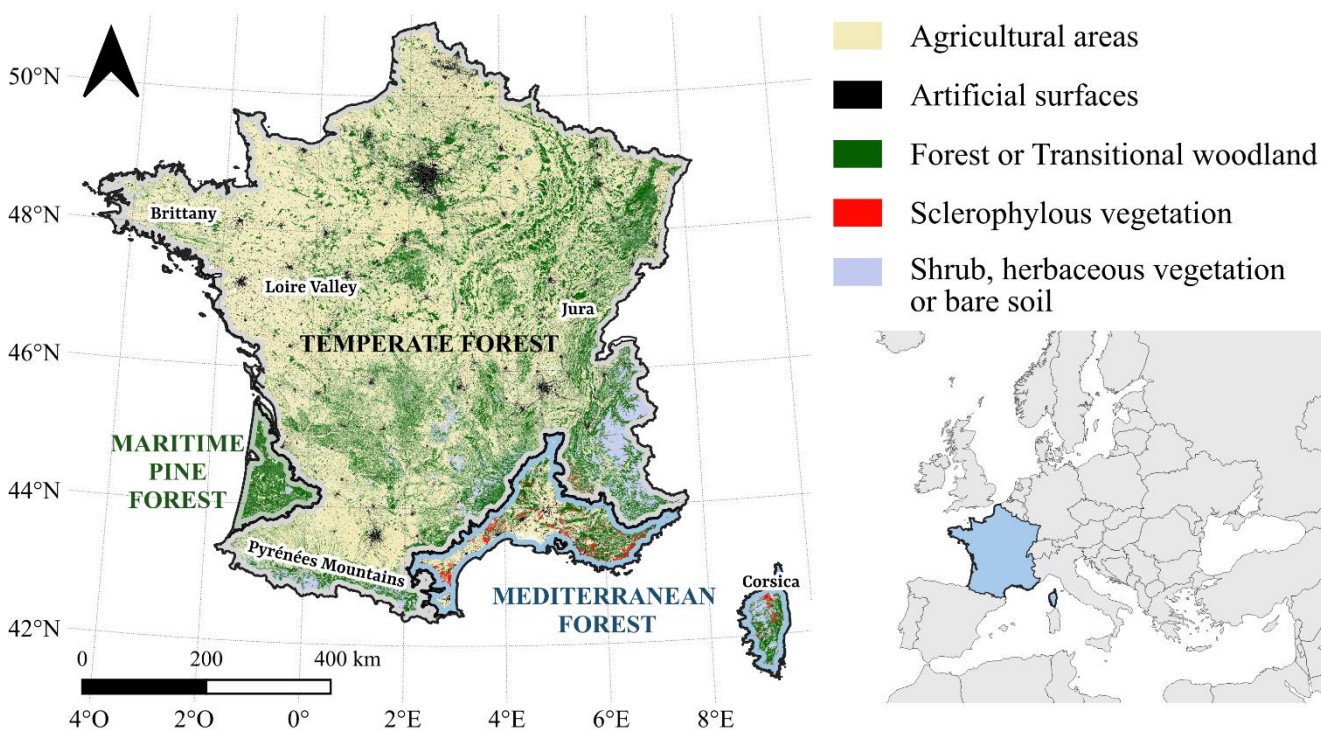


**Figure 1. Forest and vegetation cover in France from Corine Land Cover 2018 (SDES et al., 2019), with delimited major forest regions used in the study (Mediterranean forest, maritime pine forest and temperate forest). Locations discussed in the following sections (Brittany, Loire Valley, Jura and Pyrénées Mountains) are specified in black.**





**Figure 2. Three examples of height prediction within fire polygons in Atlantic Maritime Pine forests (green), Mediterranean forests (blue), and temperate forests (gray). The fire locations are indicated by the red fire icon in the first column. The second column shows Google Map imagery and the BAMTs fire polygons perimeter in red. The third column shows the 10 m canopy height, predicted with the method of Schwartz et al. (2022). Brighter colors indicate higher heights.**





**Figure 3. Workflow used in this study to estimate the Aboveground Biomass losses (AGB-L) on each fire patch. The details of each step are presented in Sect. 2.4**





**Figure 4. (a) Location of fires > 30 ha detected in France for 2006-2021 (gray) and 2022 (red) from BDIFF and BAMTs. Background colors indicate the main forest type. (b) Example of a fire polygon determined with BAMTs based on S2 imagery. The background image is from Google Maps. (c) Evolution of the burned area since 2006, colored by forest region. Data comes from BDIFF for 2006-2019 and from BAMTs for 2020-2022.**



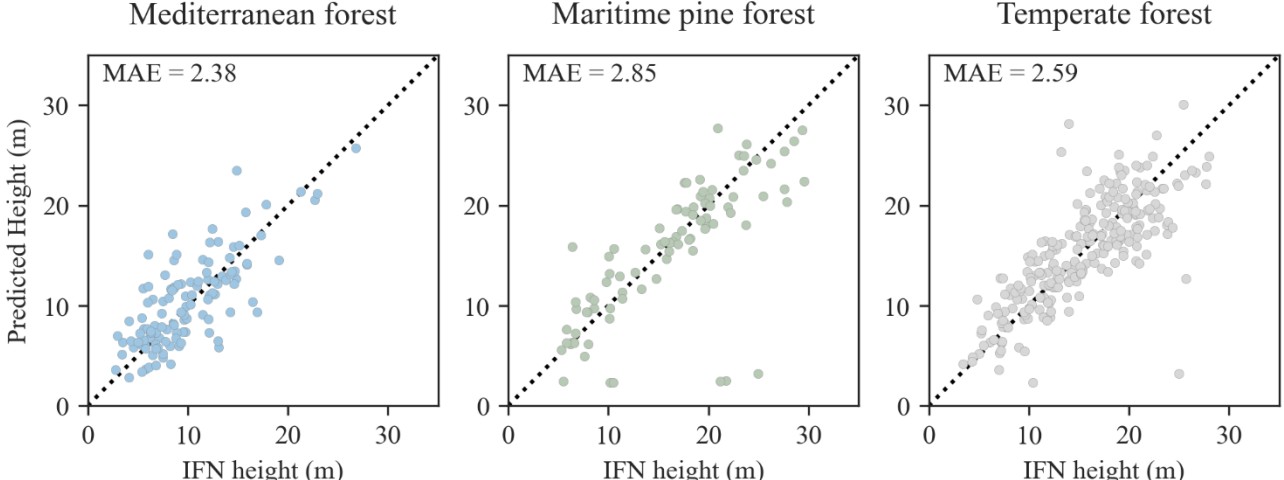

**Figure 5. Comparison of the French NFI height (2019-2021) from the plots within a 5-km buffer around fire locations and predicted values with the method developed in Schwartz et al. (2022). The black dotted lines indicate the x=y axis.**

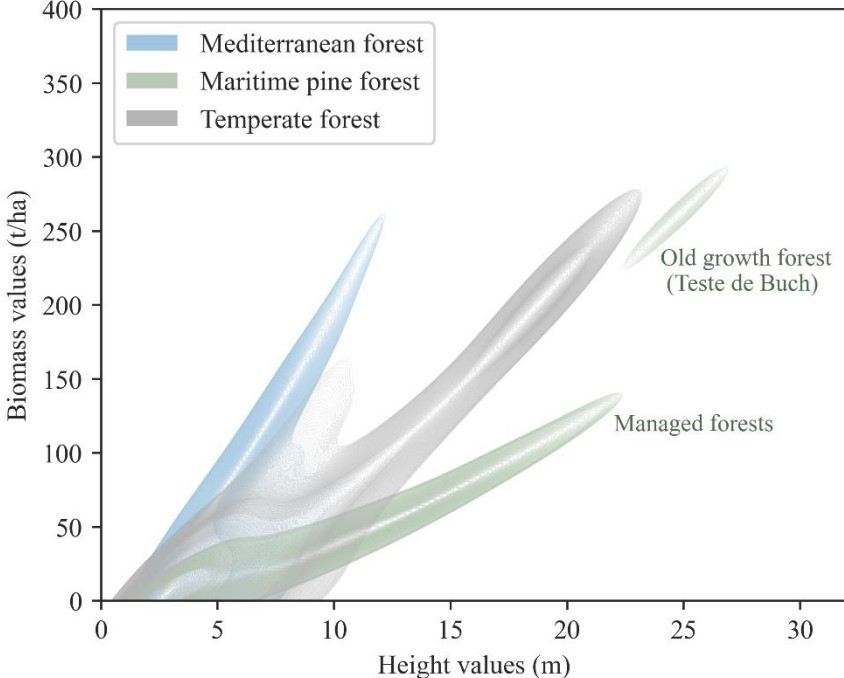


**Figure 6. Comparison of stand-level biomass values (t/ha) calculated based on the dominant tree heights (m) as described in Sect 2.4. For each forest type, we computed a contour plot of kernel density estimate that shows the distribution of all burned pixels in the Height-Biomass space.**



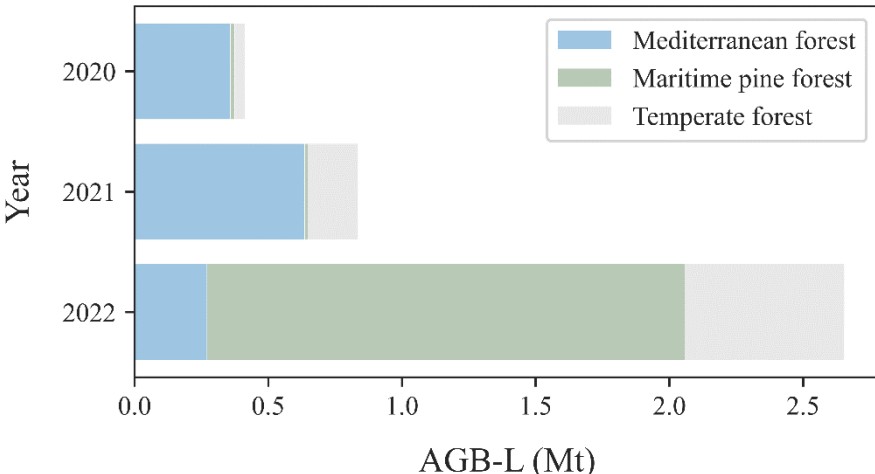

**Figure 7. Aboveground Biomass loss (AGB-L, in dry weight) by the fires in France in 2020, 2021, and 2022.**







**Figure 8. Distribution of AGB-L density (t/ ha) in Mediterranean forests (a), maritime pine forests (b), and temperate forests (c) for 2020, 2021, and 2022. The y-axis is a normalized count of the pixels with a given value of AGB. The same scale has been used for the three regions. The red dotted line represents the AGB distribution of the whole region. The y-axis has been normalized for each region.**





**Figure 9. Example of fire polygons obtained from BAMTs, EFFIS, and MCD64A1 for the fire of La Teste in an old-growth maritime pine forest with AGB-L estimations from different methods described in Sect. 2.5. The color map shows the pixel-wise estimation of AGB-L where brighter colors indicate higher AGB-L (t/ha). The figures on top of each column indicate the total AGB-L estimation for France for 2020-2022.**






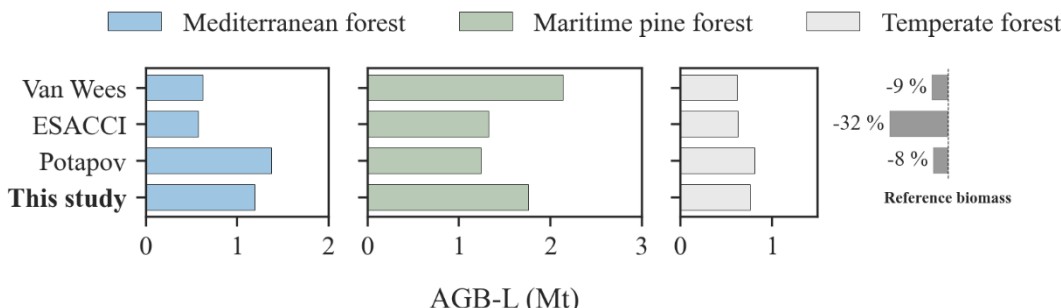

**Figure 10. Comparison of aboveground biomass loss (AGB-L, in Mt of dry weight) by fire over the 2020-2022 period estimated by our AGB-L assesment method (This study), by our method applied on Potapov tree height, by ESACCI Biomass and Van Wees 500-m model. The color represents each of the three main regions of France. The bar on the right represents the overall compared values.**






**Figure 11.** Seasonal Fire Weather Index (June-September) obtained over the period 1970-2097 for the 3 regions considered in France: Mediterranean forest, maritime pine forest and temperate forest (Copernicus Climate Change Service, 2020). The 1970-2005 data correspond to a multi-model analysis of historical data while the 2006-2097 data correspond to the RCP8.5 scenario corresponding to a constant increase of carbon emissions. The dark line corresponds to the average seasonal FWI over all pixels in the region. The seasonal FWI is the average of the daily FWI values from June 1st to September 30th. The values for the year 2022 have been recalculated from the near real-time calculation by (Field et al., 2015). 5-95% and 25-75% values intervals are also provided.






**Tables**

**Table 1. Summary of the datasets used in this study. Fire area datasets are given in the first three rows and are highlighted in red. Datasets used to estimate AGB are given in the last four rows and are highlighted in green.**

| Dataset | Resolution | Date | Units | Description | Ref | Sensors |
|---------|-----------|------|-------|-------------|-----|---------|
| BAMTS fire polygons | 10 m | 2020-2022 | Mask | Spatial extent of fires | (Bastarrika et al., 2014) | Sentinel-2 |
| EFFIS Fire polygons | refined at 20 m | 2020-2022 | Mask | Spatial extent of fires | (San-Miguel-Ayanz et al., 2012) | MODIS Sentinel-2 |
| MODIS MCD64A1 Burned areas | 500 m | 2020-2022 (monthly) | Mask | Spatial extent of fires | (Key and Benson, 2005; Giglio et al., 2018) | MODIS |
| Schwartz tree height | 10 m | 2020 | Meter | Forest Canopy Height | (Schwartz et al., 2022) | Sentinel-1, Sentinel-2, GEDI |
| Potapov Tree height | 30 m | 2019 | Meter | Forest Canopy Height (calibrated with GEDI $RH_{95}$) | (Potapov et al., 2021b) | GEDI, Landsat |
| ESA CCI Biomass AGB | 100 m | 2018 | Mg/ha | The mass, expressed as oven-dry weight of the woody parts (stem, bark, branches and twigs) of all living trees excluding stump and roots | (Santoro and Cartus, 2021) | ALOS-PALSAR Sentinel-1 |
| Van Wees 500- model AGB in burned areas (GFED framework) | 500 m | 2020 | gC/m² | Stem biomass pool | (van Wees et al., 2022) | CASA Model |

**Table 2. Area (ha) and count of fires detected between 2020 and 2022 with the method used in this study (BAMTs) and two other methods (EFFIS and MCD64A1). Fires are divided into small (<100 ha), medium (100-500 ha), and large areas (>500ha).**

| | Total | < 100 ha | 100-500 ha | >500 ha |
|---|-------|----------|-----------|---------|
| **BAMTs** | 56,053 ha 113 fires | 2,784 ha 63 fires | 8,129 ha 34 fires | 45,140 ha 16 fires |
| **EFFIS** | 62,002 ha 101 fires | 2,841 ha 51 fires | 9,373 ha 34 fires | 49,789 ha 16 fires |
| **MCD64A1** | 58,480 ha 44 fires | 278 ha 6 fires | 6,527 ha 23 fires | 51,675 ha 15 fires |
