# Peer review of "High resolution data reveal a surge of biomass loss from temperate and Atlantic pine forests, seizing the 2022 fire season distinctiveness in France"

_EGUsphere, 2023_

## Author Comment (AC1)

**Author's reply to 'Comment on egusphere-2023-529', Anonymous Referee #1, 02 Jun 2023**

This research examines historical burned area (2006-2022) in French Mediterranean, Atlantic Pine and temperature forests. 2022 was an exceptionally large fire year which led to higher than usual burned area to occur in the Atlantic Pine and temperature forests compared to more historically frequent burning in the Mediterranean systems. Burning in the old-growth Atlantic Pine and temperature forests lead to higher biomass loss than the Mediterranean forests, and by using higher resolution satellite imagery, less burned area was reported compared to EFFIS and MODIS. Additionally, Lidar based biomass estimates are combined with burned area in a novel approach.

First and foremost, the authors would like to thank referee #1 and the journal's associate editor, who agreed to supervise the peer review of this article. We would like to thank referee #1 for his comments on our study, and will do our best to respond to them in the remainder of this document.

**Comments:**

1. Line 85. This is a 0.25 degree product I believe.

Correct, this element has been corrected in the next version of the manuscript. 2. Line 134. I am a little confused how the pre and post-burn periods are defined temporally. Are NDVI, NBR and NBR2 acquired 1 year pre fire and 1 year post-fire or some other method used?

Indeed, this part of the study is not properly explained, and converges with issues raised by reviewer2. We have therefore added the following statements to the new version of the manuscript: "The fire date is provided by the first hotspot detected by FIRMS. The pre-fire period thus runs from January 1st of the year of interest to the fire date, and the post-fire period lasts from the fire date to the analysis date. The analysis date is generally several months after the fire, to guarantee a sufficient number of satellite images without cloud cover.

3. Line 136. What are the parameters in your random forest? How many trees, depth of the trees etc. How is your random forest validated? Cross validation of some sort? What are the evaluation metrics? Without knowing how well the model is performing it is hard to know if the classifier is any good. I realize you compare burned area to ERFFIS and MODIS, but the actual random forest validation metrics will be useful to include.

In the BAMT software as described in Roteta et al. 2021, we use 500 trees and the maximum tree depth ("maxNodes") was kept as default, which means that's unlimited. The evaluation is basically done through visual inspection (by comparing the burned patch with the pre- and post-composites in the background to see if it was correct), a standard for reference datasets as we mentioned (Franquesa et al.). BAMTS provides a final probability layer from the RF analysis that defines what is burned (>=50%) vs unburned. The cross validation would require 'ground reference datasets' not available, replaced by a visual inspection on each single fire. This is a semi automated method with visual checking for each single fire (not a fully automated), guaranteeing the data quality and full inspection, so we highly reduce the uncertainty own to locally varying spectral signal and affecting the performance of fully automated methods which have to be evaluated against reference datasets (as BARD dataset franquesa et al.) performed with visual checking by expert, the level of accuracy that we provided here.

4. Line 143. In general it would be better if your figures went in order, they jump from 1 to 4 here.

In order to maintain the correct sequence of figure references, we mention figure 2 here in the next version, which also shows examples of BAMTS polygons.

5. Line 144. How are you designating the forest/shrubland/pasture/grasslands? Is this an ancillary product that should be cited?

The distinction between forest and low vegetation is mentioned line 255 and 256 "all burned 10 m pixels with vegetation higher than 3 meters were treated like forests to calculate AGB-L. The pixels with vegetation shorter than 3 meters were considered to be non-forests. ". Areas considered as forest (height > 3m) acquire a biomass value according to the AGB-L method presented. Shrublands are only considered in the case of areas of sclerophyllous vegetation. This information is provided by the CORINE LC dataset. These zones are assigned a specific biomass value (10t/ha). Areas corresponding to neither forest nor sclerophyllous vegetation are considered as grassland (no distinction with pasture), and are assigned a biomass value of 4t/ha.

**6. Line 157. What type of resampling?**

We used a nearest neighbours resampling method here in order to preserve the original aspect of the 20 m resolution bands.

**7. Line 160. Which cloud mask? Citation needed.**

We used the QA60 cloud mask provided with the Sentinel-2 data in order to mask the clouds. This operation was done in Google Earth Engine as described here: https://developers.google.com/earth-engine/datasets/catalog/COPERNICUS\_S2\_SR#description

To help the reader learn more about the details of height estimation, we refer to the manuscript submitted by Schwartz et al. 2023 to the journal Earth System Science Data in the next version : Schwartz, M., Ciais, P., De Truchis, A., Chave, J., Ottlé, C., Vega, C., Wigneron, J.-P., Nicolas, M., Jouaber, S., Liu, S., Brandt, M., and Fayad, I.: FORMS: Forest Multiple Source height, wood volume, and biomass maps in France at 10 to 30 m resolution based on Sentinel-1, Sentinel-2, and GEDI data with a deep learning approach, Earth Syst. Sci. Data Discuss. [preprint], https://doi.org/10.5194/essd-2023-196, in review, 2023.

8. Line 285. Space between 66,393 and ha needed.

This has been corrected for the next version 9. Line 398. Space needed, 2022, before.

This has been corrected for the next version

---

## Author Comment (AC2)

**Author's reply to 'Comment on egusphere-2023-529', Anonymous Referee #1, 02 Jun 2023**

The authors have prepared a comprehensive and interesting manuscript about characterizing the biomass loss across a selection of fire seasons in France, through a diverse set of ecozones or forest types. The paper is generally well written, and the methods appear sound. I enjoyed the discussion at the end about how to appropriately characterize fire seasons increasingly viewed as 'exceptional' or 'unprecedented' by the media and public, with real data and analyses.

First and foremost, the authors would like to thank referee #1 and the journal's associate editor, who agreed to supervise the peer review of this article. We would like to thank referee #2 for his precise comments, which helped us to improve the content of our study. We respond to his comments in the rest of this document.

I have a few minor suggestions for clarity of the work, mostly around the methods. One topic I think requires some clarification in the text is what level of ecological fire severity is detected by the BAMTs polygons? For example, are unburned islands excluded?

BAMTs is a semi-automated method, where NBR1, NBR2 and NDVI indices are computed, based on upper canopy reflectance, and delivered as a RGB color interface to the user. Based on this visual inspection, the user defines a training area on what he considers as burnt or not. In turn, burn severity threshold is up to the user. We chose to select as much anomaly as visiually identified, by retuning the training area when evident burn RGB composites were not captured. In turn, unburned islands were conserved.

What about low-severity (no tree mortality but burning underneath)?

Since the biomass being affected by fire is all considered dead as a result of the burn, it would be helpful to better understand whether the polygons produced from the BAMT method are including only high severity, or some sort of mix (i.e. surface fires, or 'underburning'), as this would be a source of error, and lead to overstating biomass loss, if mixed and low severity fire is included within the burned areas, and this would need to be clearly acknowledged. This difference could also account for some of the differences in area burned estimates with other fire mapping products, which are described as overestimations by the other products.

The pixel reflectance is driven by the upper vegetation layer. then dNBR has been shown to hardly capture understorey fires, what ever the sensor (Roy et al. 2006, Morton et al. 2013). In turn, our method does not capture understorey fires, but high to mid severity fires affecting the canopy. This 'weakness' in capturing understorey fire is similar for all sensors as they use a similar approach using the dNBR index. Mismatches in burned areas is then mostly a consequence of pixel resolution, and algorithm thresholds to consider a partially-burned pixel as fully burned. Uncertainty indices are provided in global remote sensing of burned area, potentially reducing their BA, but they are hardly used in global BA estimates and Emission.
Only recent advances using Lidar may provide new information for these kind of fires for now (East et al. 2023).  In turn ground fires under the canopy were not captured, so we could use our burned area specifically focused on tree biomass affected.
We'll provide further details and shortcomings regarding this issue in the updated version. We agree this would be of interest for the reader.

Roy, D.P.; Boschetti, L.; Trigg, S.N. Remote Sensing of Fire Severity: Assessing the Performance of the Normalized Burn Ratio. IEEE Geosci. Remote Sens. Lett. **2006**, 3, 112–116.

East, A.; Hansen, A.; Armenteras, D.; Jantz, P.; Roberts, D.W. Measuring Understory Fire Effects from Space: Canopy Change in Response to Tropical Understory Fire and What This Means for Applications of GEDI to Tropical Forest Fire. *Remote Sens.* **2023**, *15*, 696. https://doi.org/10.3390/rs15030696

Morton DC, Le Page Y, DeFries R, Collatz GJ, Hurtt GC. Understorey fire frequency and the fate of burned forests in southern Amazonia. Philos Trans R Soc Lond B Biol Sci. 2013 Apr 22;368(1619):20120163. doi: 10.1098/rstb.2012.0163. PMID: 23610169; PMCID: PMC3638429.

Title: possibly revise "seizing" to another word? Maybe "characterizing" or "contextualizing", instead?

We agree we could use a better terminology here. We might use 'recontextualizing'.

Line 17: hyphenate fire-prone?

This has been corrected for the next version

Line 62: replace has with have.

This has been corrected for the next version

Line 73: AGB-L is defined as the acronym, but for most of the rest of the introduction section AGB is used alone or Loss is spelled out. I also feel that this definition of AGB-L is very important. This is not combustion but rather is combustion combined with mortality of live biomass. This is why I feel that some clarification about whether there is any mixed-mortality wildfire captured in the burned polygons is needed.

To take your comment into account, we've made the use of the acronym AGB-L more present in the rest of the document. We also modified the definition of AGB-L to make it more comprehensive : "AGB-Lis defined as all direct and indirect potential biomass losses due to fire. These potential losses then include all the biomass exposed to fire, leading to either the combustion of the plant material during the fire, resulting in the release of gases and aerosols, or the formation of standing and ground dead wood, which is then decomposed or harvested by forest managers. This definition refers to the concept of potential loss used in fire risk assessments (Chuvieco et al. 2023)"

Line 134: Please define the range of dates considered for pre-fire/post-fire. Were these 'initial' assessments (immediately before and after fire)? Multi-year (extended assessment)? Either is okay but the methods are not replicable without these details.

Indeed, this part of the study is not properly explained. We have therefore added the following statements to the new version of the manuscript:
"The fire date is provided by the first hotspot detected by FIRMS. The pre-fire period thus runs from January 1st of the year of interest to the fire date, and the post-fire period lasts from the fire date to the analysis date. The analysis date is generally several months after the fire, to guarantee a sufficient number of satellite images without cloud cover.

Line 137: Can the authors add to the text to explain how the BAMTs determines burned/unburned? Is the training data supplied by the user of the tool, and specific to the region, or is automatically supplied by the tool? What area or region is the training data from?

Regarding the previous concerns about fire severity, that we'll develop further, we'll have to include this technical step, where the user defines and tunes his own training area according to the visual inspection of the composite of three indices.

Line 190: How many NFI plots were used for each patch/model parameter p? Was there a minimum number of plots used?  What was the range?

 In fact, we have not indicated the characteristics of this parameter. The buffer of 5 km around the fire allowed us to consider a sufficient number of plots. The minimum plot value is 5, and the maximum is 278, with a median of 41. We have added this sentence on line 195 of the next version of the manuscript :
"This buffer made it possible to rely on a sufficient number of plots ranging from a minimum of 5 plots to 278 (median = 41)."

Line 365: Although I recognize that the old-growth forests are likely the highest biomass stands on the landscape, I'm not sure that the differences between the two distributions fully support the statement "affected a higher proportion of old-growth forests than were available to burn". I don't think that's possible, since even if it affected 100% of them that is still the maximum that would be available to burn. A couple suggestions to address this would be to introduce statistical tests that compare distributions to determine whether they are significantly different (e.g., K-S test), which would then support the authors saying something like "burned stands that had a significantly higher biomass than was typical for the region". Alternatively, or additionally, another option would be to use something like a chi-square test or likelihood ratio tests of the spatial data to compare how much old growth was available on the landscape, and whether they preferentially burned, relative to their availability.

Thank you for this particularly pertinent comment, which adds robustness to what we're saying. We therefore decided to add the Kolmogorov-Smirnov test to our Materials and methods section, and to add the output of this test to the Results section :
« 2.6 Statistical test
To establish whether a difference exists between two distributions, we used the Kolmogorov-Smirnov test (KS-test). This test compares the parameters of two distributions (mean and variances) to conclude whether their difference is significant. A p-value below the 0.05 threshold indicates a significant difference between the two distributions. To perform this, we used the ks.test function in the "stat" set of the R program."
We indeed obtained significant differences in the distribution of affected and living biomass.

Line 420: Typo "three" instead of tree.

This has been corrected for the next version

I found the figure caption for Figure 10 a bit hard to follow, specifically what the reference product being compared to was and what comparison was being made.

We have taken your comment into account and modified the legend as follows in the next version:
« Figure 10. Comparison of aboveground biomass loss (AGB-L, in Mt of dry weight) by all considered fires over the 2020-2022 period estimated by our AGB-L assesment method (This study), by our method applied on Potapov tree height, by ESACCI Biomass and Van Wees 500-m model. The color represents each of the three main regions of France. The bars on the right correspond to the difference in AGB-L over all three regions (in percentage) between our method and the other data sources. »